# Silver and Samaria-Doped Ceria (Ag-SDC) Cermet Cathode for Low-Temperature Solid Oxide Fuel Cells

**DOI:** 10.3390/nano13050886

**Published:** 2023-02-27

**Authors:** Davin Jeong, Yonghyun Lim, Hyeontaek Kim, Yongchan Park, Soonwook Hong

**Affiliations:** 1Department of Mechanical Engineering, Chonnam National University, 77 Yongbong-ro, Buk-gu, Gwangju 61186, Republic of Korea; 2Department of Mechanical Engineering, Stanford University, Stanford, CA 94305, USA

**Keywords:** silver (Ag), samaria-doped ceria (SDC), co-sputtering, solid oxide fuel cell (SOFC), oxygen reduction reaction (ORR)

## Abstract

This study demonstrated a silver (Ag) and samarium-doped ceria (SDC) mixed ceramic and metal composite (i.e., cermet) as a cathode for low-temperature solid oxide fuel cells (LT-SOFCs). Introducing the Ag-SDC cermet cathode for LT-SOFCs revealed that the ratio between Ag and SDC, which is a crucial factor for catalytic reactions, can be tuned by the co-sputtering process, resulting in enhanced triple phase boundary (TPB) density in the nanostructure. Ag-SDC cermet not only successfully performed as a cathode to increase the performance of LT-SOFCs by decreasing polarization resistance but also exceeded the catalytic activity of platinum (Pt) due to the improved oxygen reduction reaction (ORR). It was also found that less than half of Ag content was effective to increase TPB density, preventing oxidation of the Ag surface as well.

## 1. Introduction

Solid oxide fuel cells (SOFCs) have emerged as promising renewable energy conversion devices due to their high efficiency, zero pollutant emissions, and fuel flexibility [1,2,3,4]. Despite extensive research over the decades, there are still issues to overcome for commercialization. SOFCs are generally operated at 800–1000 °C, which causes the thermal degradation of the catalyst, material selection, and seal integrity [3,4,5,6]. Therefore, extensive studies have been carried out to reduce the operating temperature of SOFCs to a range of 400–600 °C [7,8,9,10]. Although low-temperature SOFCs (LT-SOFCs) have posed new challenges related to reduced ionic conductivity (i.e., Ohmic loss), recent advancements in thin-film fabrication techniques have allowed for the production of ceramic electrolytes thinner than 1 μm, leading to a significant reduction of Ohmic loss [5,11,12]. However, in the lower temperature region, polarization loss at the cathode–electrolyte interface becomes a more dominant factor decreasing the performance of LT-SOFCs than Ohmic loss. This phenomenon is primarily attributed to the decreased temperature causing a sluggish surface oxygen reaction rate. Therefore, many efforts have been made to develop materials that have high catalytic activity to increase the surface oxygen reduction reaction (ORR) [13,14,15,16,17].

The most widely used cathode materials for SOFCs are lanthanum-based perovskite materials, such as La_1−*x*_Sr*_x_*MnO_3_, La_1−*x*_Sr*_x_*CoO_3_, and La_1−*x*_Sr*_x_*Co_1−*y*_Fe*_y_*O_3_ [18,19,20]. Since the activation energy of the above materials is quite high (<1.5 eV), polarization loss substantially increases as the operating temperature of SOFCs is lowered [21]. Hence, platinum (Pt) and its alloys, which are well known for their superior catalytic activity compared to all other materials, have been utilized as cathode materials to increase ORR kinetics at low temperatures [11,15,16]. Despite the advantages of Pt-based catalysts, their high cost and scarcity make them an impractical choice as catalysts.

One of the alternatives to Pt, silver (Ag), has been suggested as a cathode material because of its high ORR activity, high oxygen diffusion rate and relatively low price. From this point of view, many previous studies reported that Ag based alloy can achieve high catalytic activity for ORR kinetics comparable to Pt catalyst [22,23,24], and even higher electrode activity can be obtained by combining with oxygen ionic conductor to extend triple phase boundary, which is the reaction site and physical contact point among Ag, ionic conductor, and oxygen gas [25]. For example, Kamlungsua et al. demonstrated that the Ag and samarium-doped ceria (SDC) core-shell based cathode showed improved surface oxygen adsorption/dissociation kinetics by enlarging the TPB density [26]. Kim et al. also demonstrated that the Ag cathode encapsulated by YSZ ionic conductor showed improved polarization kinetics compared to the Pt catalytic electrode [27]. However, a recent investigation of Ag cathodes reported that the Ag cathode’s TPB can be decreased with agglomeration and become severely degraded by the formation of an oxide at the surface as well, resulting in reduced catalytic activity even at low temperatures [25,28]. These issues directly decrease the performance of LT-SOFCs.

In this study, we demonstrate the Ag-SDC metal-ceramic (i.e., cermet) composite cathode for LT-SOFCs revealing higher electrochemical performance than Pt catalytic cathode. SDC is utilized to make cermet cathode with Ag, based on its high ionic conductivity among the doped-ceria. The employed cermet cathode successfully demonstrates the increase of TPB density by optimizing the composition between Ag and SDC, ensuring the high performance of LT-SOFCs at 400 °C. It is also confirmed that the Ag-SDC cermet cathode shows mitigation of the agglomeration and passivation of the Ag surface from the oxidation at the operating temperature. We believe that the results presented herein will help to fabricate low-cost and high performance cathode for LT-SOFCs and facilitate their commercialization.

## 2. Materials and Methods

### 2.1. Sample Preparation

Polycrystalline silicon wafer (110) substrates with dimensions of 1 cm × 1 cm × 0.5 mm were used to investigate the morphology, crystallinity, and composition of the cathodes fabricated with different ratios of Ag and SDC. We used commercially available 8 mol % YSZ substrates with dimensions of 1 cm × 1 cm × 0.2 mm (MTI Corporation, Richmond, CA, USA) to fabricate the fuel cells for the electrochemical analysis of Ag-SDC cermet cathodes.

We employed a co-sputtering system (NEO, JA Innovation, Yongin, Republic of Korea) to deposit the Ag-SDC cermet cathodes. An Ag target (2-inch diameter, 2-mm thickness, 99.9% purity) and an SDC target composed of Sm_0.1_Ce_0.9_O_2–*x*_ with the same dimensions (RND Korea Corp., Gwangmyeong, Republic of Korea) were utilized to fabricate the various compositions of Ag-SDC cermet cathodes on YSZ substrates. The base pressure of the sputtering chamber was 5 × 10^−^^5^ Pa. The targets were placed 10 cm away from the substrate which was mounted on a substrate holder. The co-sputtering process was conducted under 10 Pa of working pressure with an argon gas flow rate of 20 sccm at room temperature. The Ag-SDC cermet cathodes with four different ratios of Ag and SDC were fabricated by varying the power of each direct current (DC) and radio frequency (RF): DC powers of 20, 60, and 100 W were applied for the deposition of Ag, and RF powers of 50 and 80 W were applied for the deposition of SDC (Figure 1). The thickness of all the fabricated Ag-SDC cermet cathodes was kept at about 100 nm, and the area of the cathodes was 1 mm × 1 mm. Consequently, the Pt was deposited as an anode on the YSZ substrate with a DC power of 100 W under the same sputtering conditions of Ag-SDC deposition.

### 2.2. Fuel Cell Characterization

Field-emission scanning electron microscopy (FE-SEM, S-4700, Hitachi, Tokyo, Japan) and atomic force microscopy (AFM, XE-100, Park Systems, Suwon, Republic of Korea) were employed to observe the surface morphology of the Ag-SDC cermet thin films with different ratios. The crystalline phases of the Ag-SDC cermet cathodes were analyzed by X-ray diffraction (XRD, X’Pert pro MPD, PANalytical, Worcestershire, UK). To verify the composition of Ag-SDC, X-ray photoelectron microscopy (XPS, K-Alpha, Thermo Fisher Scientific, Waltham, MA, USA) and Al Kα monochromatic radiation were used.

The performance of fuel cells with Ag-SDC cermet cathodes was measured using a homemade probing system with a temperature-controllable substrate heater supplying a constant temperature of 400 °C. The polarization curves (i.e., current–voltage–power density behavior) of the Ag-SDC cermet-coated fuel cells were evaluated via an electrochemical analyzer (Interface 1010E, Gamry Instruments Co., Ltd., Warminster, PA, USA). Electrochemical impedance spectroscopy (EIS) was also conducted with the same apparatus in the range between 1 MHz and 0.5 Hz. The equivalent circuit models of ZView (Scribner Associates Inc., Southern Pines, NC, USA) were applied to analyze the measured data.

## 3. Results and Discussion

To obtain each different chemical composition of the Ag-SDC cermet cathodes, the DC sputtering power of Ag and the RF sputtering power of SDC were manipulated. Table 1 shows the XPS atomic percentage results of the fabricated Ag-SDC thin films. It is clear that the atomic composition of Ag in the Ag-SDC films was proportional to the increased sputtering power of Ag, while it was inversely proportional to the increased sputtering power of SDC. Therefore, it was speculated that these varied ranges of Ag and SDC atomic ratios for Ag-SDC cermet cathodes may directly correlate to the performance of SOFCs with different TPB densities inside of their nanostructures. The XPS spectra of the Ag-SDC cermet were also presented in Figure 2 for further chemical information of each. The peaks of Sm3d and Ce3d were clearly shown at relatively lowered Ag content (i.e., Ag20-SDC50 and Ag20-SDC80 samples), while the strong peaks of Ag3d were observed in high content of Ag samples (i.e., Ag60-SDC50 and Ag100-SDC50 samples). This phenomenon revealed that higher DC and RF powers during the sputtering process led to greater contents of each element as confirmed with XPS results.

Figure 3 indicates the growth rate of the Ag-SDC cermet thin films in various conditions. The growth rates were calculated with the films thickness by dividing into deposition time. It was revealed that the deposition rate was significantly elevated when the DC power of Ag was increased from 20 W to 60 W and 100 W, while altering the RF power of SDC had less of an effect on the growth rate. In this regard, we confirmed that the key factor influencing the deposition rate was DC power during the sputtering process rather than RF power, considering the deposition rate of both the Ag60-SDC50 and Ag100-SDC50 cermet cathodes. Moreover, the above two sputtering conditions also showed high concentrations of Ag content, as shown in Table 1, leading to almost Ag metal-phase cathodes. On the other hand, the sputtering conditions of both Ag20-SDC50 and Ag20-SDC80, in which a low DC power of Ag was applied, exhibited almost the same deposition rates. Therefore, considering both the XPS results and calculated deposition rates, the lowered sputtering power of the DC source is effective to fabricate Ag-SDC cermet cathodes with a balanced atomic ratio between Ag and SDC.

The nanostructures of Ag-SDC cermet cathodes were investigated with SEM images (Figure 4). We found that the morphology of all the Ag-SDC thin films presented almost identical grain structures except the film deposited with Ag20-SDC50. The Ag20-SDC50 thin film showed a vague morphology with particles on the surface. This nanostructure can be attributed to the relatively lower DC and RF sputtering powers, which may reduce surface adatom mobility and restrain the grain growth in the sputtering process [29,30]. Moreover, it was also reported that this nanostructural tendency can be more remarkable in Ag-ceramic composites because the Ag easily segregates to the grain boundaries and hinders the grain growth during the deposition [31]. The aggregation of Ag particles on the surface of the film was detected as a result of reducing the surface energy of metallic particles during the film growth (Figure 4c), which is in line with previous reports [32,33]. However, more interestingly, the higher RF power condition of Ag20-SDC80 than Ag20-SDC50 showed well-developed grains (Figure 4d). It was assumed that the sufficient surface adatom mobility allowed the adatoms to migrate to energetically favorable locations to minimize their surface energy, resulting in grain development. Thus, no Ag aggregation was observed in the Ag20-SDC80 thin film, implying that there were stabilized Ag atoms inside of the film structure. The AFM results showed the same tendency of surface topography for all the Ag-SDC cermet cathodes (Figure 5). The cermet film deposited with the Ag20-SDC50 sputtering condition also showed nanoparticles on the surface, while the other films revealed relatively developed grains in their nanostructures.

The crystallinity of the deposited Ag-SDC cermet cathodes was analyzed by XRD, and the diffraction spectra are shown in Figure 6. The XRD spectrum of the Ag20-SDC50 thin film demonstrated amorphous characteristics without any perceptible peaks. This phenomenon is associated with the insufficient mobility of the surface adatoms and the deterioration of the continuous film growth owing to the formation of Ag nanoparticles, as visually confirmed in the SEM and AFM results (Figure 4 and Figure 5). For the other sputtering conditions of Ag-SDC thin films, however, the peaks related with the Ag crystalline phase were notably observed in the XRD spectra that were preferentially along the (111) direction. There were no distinguishable peaks correlated with SDC, which implies that the restriction of crystalline development for SDC occurred during the deposition. In the silver and oxide composite film, the smaller grains of silver atoms tend to easily detach and diffuse to larger grains in the matrix to reduce the total surface area, and thus such morphological characteristic can lead to the formation of amorphous oxide and silver crystalline mixture [33]. Therefore, we confirmed that the development of the crystalline phase in Ag-SDC cermet thin films mainly came from the grain development of the Ag element by the agreement with the XPS results.

To evaluate the electrochemical performance of the Ag-SDC cermet cathodes, polarization behaviors of the fuel cells with various Ag-SDC cermet cathodes and a reference Pt cathode were measured, as shown in Figure 7. The peak power densities of the fuel cells to which Ag20-SDC80, Ag20-SDC50, Ag60-SDC50, and Ag100-SDC50 cermet cathodes were applied were 1.77, 1.53, 1.29, and 0.93 mW/cm^2^ at 400 °C, respectively, while the performance of the fuel cell with only the Pt cathode was 1.47 mW/cm^2^ at the same temperature. This result indicated that the performance of the fuel cells with Ag60-SDC50 and Ag100-SDC50 cermet cathodes was poor, whereas the fuel cells with Ag-SDC cermet cathodes with lowered Ag content (i.e., Ag20-SDC50 and Ag20-SDC80) outperformed the fuel cells with Pt cathodes. The peak power density of the fuel cell with Ag20-SDC80 cathode (1.77 mW/cm^2^) is about 20% higher than that with the Pt cathode (1.47 mW/cm^2^) at 400 °C, which is similar enhancement achieved in previous report [25]. The developed grains and crystalline phase of the Ag element for both the Ag60-SDC50 and Ag100-SDC50 cermet cathodes did not significantly affect the performance of fuel cells. Thus, it was suggested that the factor with the greatest contribution to the performance of fuel cells was the TPB density of Ag-SDC cermet cathodes with balanced Ag and SDC content. It is also noted that the increased TPB density inside of cermet cathodes with the Ag20-SDC50 and Ag20-SDC80 can substitute Pt cathode by enhancing ORR kinetics in LT-SOFCs. In addition, the performance of the fuel cells with Ag20-SDC50 cermet cathodes also exceeded that of Pt-coated fuel cells even though Ag nanoparticles existed on the surface.

To clarify the evidence of the enhanced performance of fuel cells with Ag-SDC cermet cathodes, EIS analysis was conducted, as shown in Figure 8. Two main semicircles were clearly observed in the spectra for all fuel cells we made. It is generally accepted that the radius of the first semicircle intercepted with the *x*-axis represents the Ohmic resistance, while that of the second semicircle is correlated with the polarization resistance. According to the observation of the first semicircles, the EIS spectra of the fuel cells with Ag20-SDC80 and Ag20-SDC50 indicated almost the same radii as that of the Pt-coated fuel cell. However, the Ohmic resistance of both the Ag60-SDC50 and Ag100-SDC50 cermet-coated fuel cells was increased. This result revealed that the higher Ag content of the Ag-SDC cermet cathodes increased the oxidation of Ag elements at the surface of cathodes, resulting in decreased electrical conductivity. In contrast, the polarization resistance significantly differed among the fuel cells. The calculated polarization resistance values were 116.2, 57, 128, 315, and 133.4 Ω·cm^2^ for the Ag20-SDC50, Ag20-SDC80, Ag60-SDC50, Ag100-SDC50, and Pt cathodes, respectively. Compared to the Pt cathode-coated fuel cell, the polarization resistance of the fuel cell with the Ag20-SDC80 cermet cathode was significantly improved. It was regarded that the cermet cathode can achieve extended TPB density throughout the entire volume of the layer, while the Pt catalytic layer has a limited reaction area (only at the electrolyte/electrode interface). It was also confirmed that the polarization resistance of the Ag20-SDC50-coated fuel cell was greater than that of the Ag20-SDC80-coated fuel cell but almost the same as that of the Pt-coated fuel cell. It is clear evidence that the main reason for the enhanced fuel cell performance is the increased ORR kinetics with well percolated Ag and SDC both increasing the TPB density of the cermet cathode. Therefore, a superior cathode structure can be accomplished with Ag-SDC cermet instead of precious Pt material.

## 4. Conclusions

Pt is considered one of the most favorable materials as a cathode for SOFCs in low-temperature operation due to its superior catalytic activity compared to lanthanum-based perovskite materials. However, considering the price and scarcity of Pt, Ag has emerged as a promising candidate cathode material owing to its highly active ORR and relatively low price. While Ag alloys have been extensively studied to alter the Pt cathode, it has been found that Ag is thermally vulnerable to oxidation at the surface and agglomeration, resulting in reduced TPB density. We successfully demonstrated that an Ag-based cermet cathode with SDC can not only mitigate the above thermal issues but also significantly enhance the ORR by increasing the TPB density inside of the cathode nanostructure. It is expected that this study’s findings will lead Ag-SDC cermet cathodes to be applied to other energy-conversion applications, accelerating commercialization.

## Figures and Tables

**Figure 1 nanomaterials-13-00886-f001:**
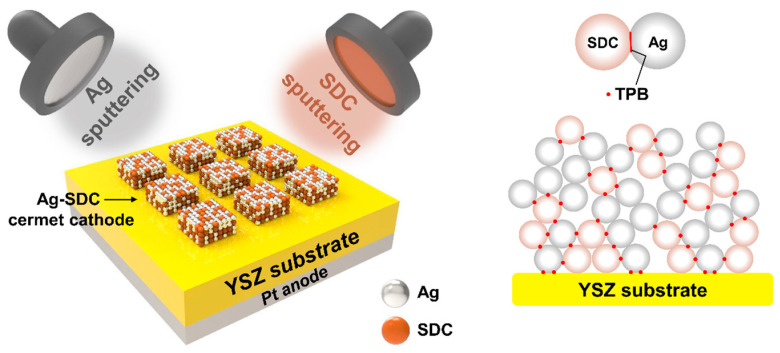
Co-sputtering process for Ag-SDC cermet cathodes.

**Figure 2 nanomaterials-13-00886-f002:**
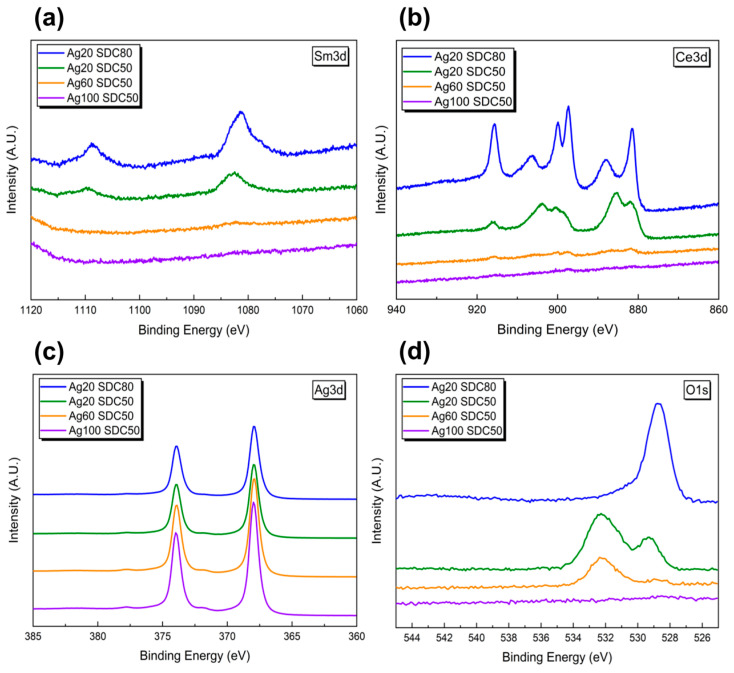
The XPS spectra of the (**a**) Sm3d, (**b**) Ce3d, (**c**) Ag3d and (**d**) O1s for various Ag-SDC cermet cathodes.

**Figure 3 nanomaterials-13-00886-f003:**
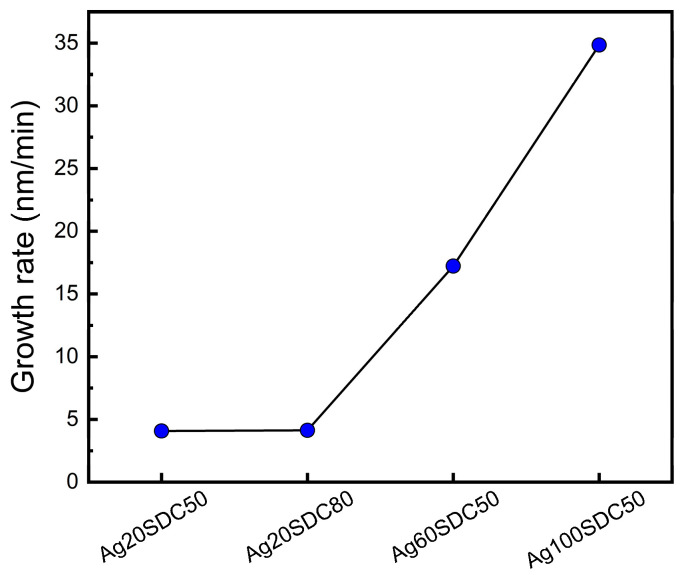
Growth rate of Ag-SDC cermet cathodes with various sputtering power conditions.

**Figure 4 nanomaterials-13-00886-f004:**
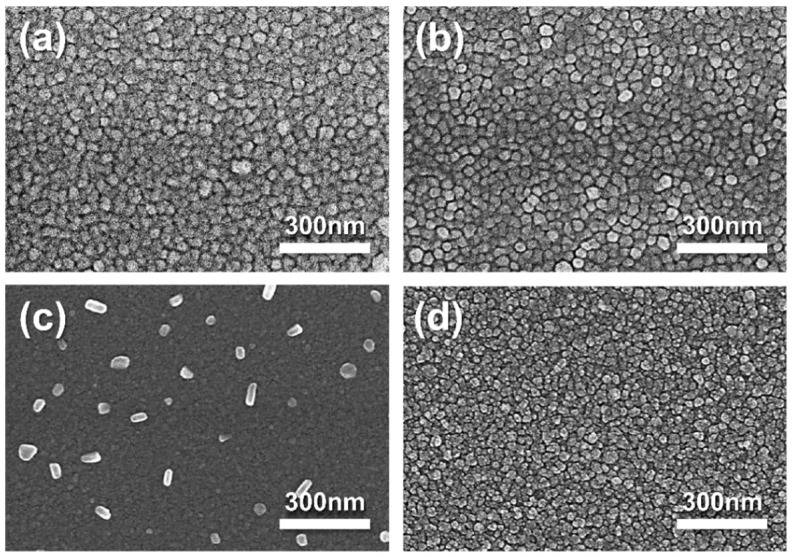
SEM micrographs of 100-nm-thick Ag-SDC cermet cathodes deposited under DC and RF powers of (**a**) Ag100-SDC50, (**b**) Ag60-SDC50, (**c**) Ag20-SDC50, and (**d**) Ag20-SDC80 on YSZ substrate.

**Figure 5 nanomaterials-13-00886-f005:**
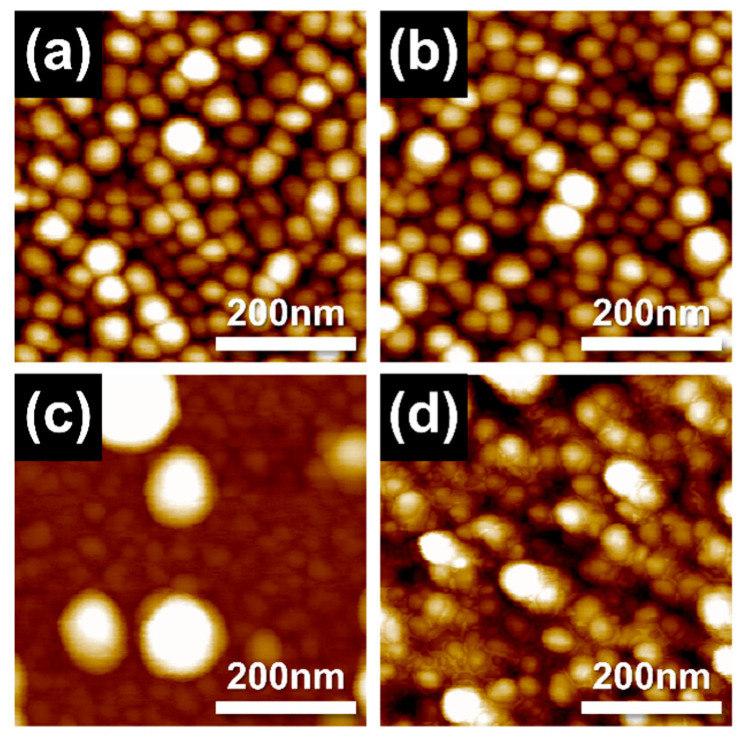
AFM topography of 100-nm-thick Ag-SDC cermet cathodes deposited under DC and RF powers of (**a**) Ag100-SDC50, (**b**) Ag60-SDC50, (**c**) Ag20-SDC50, and (**d**) Ag20-SDC80 on YSZ substrate.

**Figure 6 nanomaterials-13-00886-f006:**
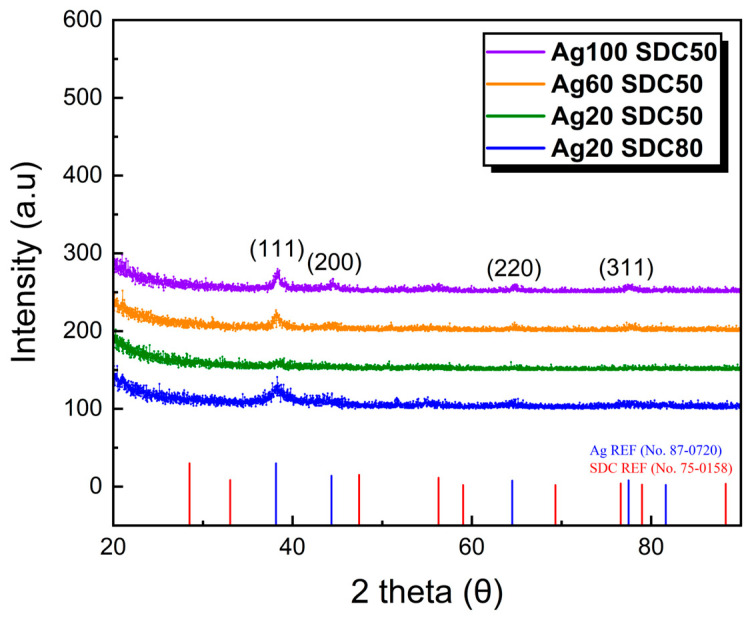
XRD spectra of 100-nm-thick Ag-SDC cermet cathodes fabricated on silicon substrate using various DC and RF powers.

**Figure 7 nanomaterials-13-00886-f007:**
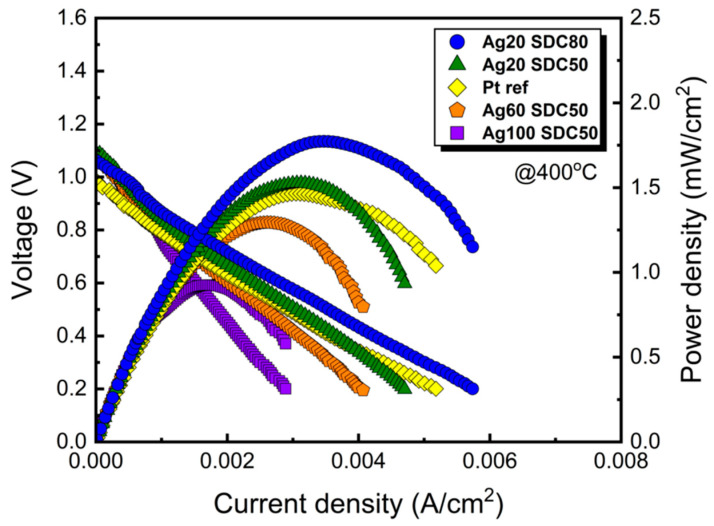
Polarization curves for fuel cells with different DC and RF powers of Ag-SDC cermet cathodes coated on YSZ substrates. All the fuel cells were measured at 400 °C.

**Figure 8 nanomaterials-13-00886-f008:**
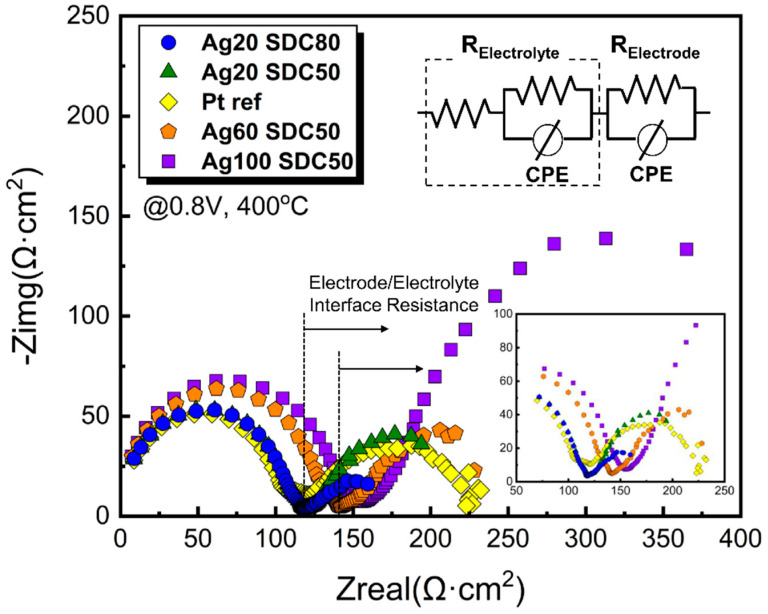
EIS spectra of Ag-SDC cermet cathodes coated on YSZ substrates with various DC and RF powers. The inset shows the low-frequency region of EIS data.

**Table 1 nanomaterials-13-00886-t001:** XPS results of atomic percentages of Ag-SDC cermet with different ratios of Ag and SDC.

**Power of Ag (DC)**	20 W	20 W	60 W	100 W
**Power of SDC (RF)**	50 W	80 W	50 W	50 W
Ag3d (at%)	65.55	38.24	88.86	99.79
Ce3d (at%)	5.34	11.78	0.87	0.21
Sm3d (at%)	0.81	1.45	0	0
O1s (at%)	28.3	48.53	10.27	0

## Data Availability

The data in this study are available upon request from the corresponding author.

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
