# Peer review of "Silver and Samaria-Doped Ceria (Ag-SDC) Cermet Cathode for Low-Temperature Solid Oxide Fuel Cells"

_nanomaterials, 2023, doi:10.3390/nano13050886_

Round 1

Reviewer 1 Report

1.        There is no obvious characteristic peak in the XRD pattern in Figure 5. The author should give a more convincing explanation and provide reliable references.

2.        In the introduction, the author should add more relevant literature, the purpose of the study is not clear enough.

3.        The citation format of the document is incorrect. For example, [32-33] should be changed to [32, 33].

4.        Large pieces of literature references should be avoided, such as [11-16], [21-25], etc.

5.        Please update the references. The latest research results of the last five years should be quoted more.

6.        In the discussion of the experimental results, there is a lack of comparison with other literature results to highlight the research significance of this work.

Reviewer 2 Report

The authors prepared and tested different co-sputtered silver (Ag) and samarium-doped ceria (SDC) cermets as possible cathodes for LT-SOFCs, demonstrating to have superior electrochemical performances and higher TPB densities than pure Pt cathodes. However, there are some major issues to be addressed by the authors prior publishing their manuscript to Materials:

Introduction should be expanded to better highlights possible alternatives to Pt cathodes and recent research trends. There are many recent papers that could be cited within the introduction, even in terms of Ag-SDC composites ("Kamlungsua, K., Lee, T. H., Lee, S., Su, P. C., & Yoon, Y. J. (2021). Inkjet-printed Ag@ SDC core-shell nanoparticles as a high-performance cathode for low-temperature solid oxide fuel cells. International Journal of Hydrogen Energy46(60), 30853-30860");

Authors should add a figure and a related discussion on the obtained XPS results (rather than just a table with the atomic percentages of the different elements);

SEM of the various samples at higher magnifications should be added within the manuscript, and the Ag20W-SDC50W different morphology should be better justified;

About the XRD results, you should use diffraction patterns starting from at least 20 °, as the main amorphous halo fo the SDC is generally visible around 26 - 30 °;

You should uniform sample labelling within the manuscript and figure captions (i.e. you alternatively use Ag20W-SDC50W and Ag20-SDC50);

You should imrpove the discussion of the obtained electrochemical results, trying to better explain why Ag20W-SDC50W and Ag20W-SDC80W electrodes could behave better than a Pt electrode in LT-SOFCs.

Author Response

Please see the attachment. Thanks for your careful revision.

Reviewer 3 Report

The paper entitled “Silver and Samaria-doped Ceria (Ag-SDC) Cermet Cathode for Low-Temperature Solid Oxide Fuel Cells” is devoted to the development of electrodes for application in low-temperature SOFCs manufactured by a sputtering process. This topic is of potential interest to the readership of the journal. However, the manuscript in the current version can not be published. No novelty and important results are presented. Obtained power density values about 1.7 mW/cm2 are incredibly small and do not have practical significance.  The methodology of the electrode preparation should be expanded with details of the sputtering process; abbreviations (DC and RF) should be explained.  Did the authors perform a heat treatment of the electrodes after sputtering? According to XRD data there is no SDC phase in the electrodes at all, it is even hard to name it “cermet” and to expect enhanced ORR and electrochemical activity. Authors claimed that the TPB is expanded, but three phases do not exist in the studied samples. Difference in the activity between Pt and “Ag-cermet” cathodes is 0.25 mW/cm2, it is negligible small and can not be considered as “improved performance”. 

Author Response

(The authors gave the same response as above.)

Round 2

Reviewer 2 Report

The authors well addressed all the proposed issues, and the overall quality of the manuscript has been significantly improved. Therefore, I recommend the revised manuscript to be published in Nanomaterials. 

Author Response

Thanks for your kind suggestion for pulication of this paper. All authors are really appreciate for improvement of manuscript. 

Reviewer 3 Report

The authors have answered to all questions and revised the manuscript. 

However, there is still no evidence of the existence of SDC phase in the studied cathodes. The improvement of TPB when composite electrodes are applied is obvious and there is nothing new here. Additionally, one of the questions not raised in the paper is the long-term performance of such cathodes. It is obvious that due to the absence of preliminary heat-treatment the electrode should sinter under working conditions of the fuel cell leading to changes in the microstructure and phase composition and, as a consequence, changes in power density values. 
